# OpenReview forum: "It's a TRAP! Task-Redirecting Agent Persuasion Benchmark for Web Agents"
_ICLR.cc/2026/Conference — Submitted to ICLR 2026_

### Official Review · Reviewer_4gAY · 2025-11-01

**Soundness:** 3
**Presentation:** 3
**Contribution:** 3
**Rating:** 6
**Confidence:** 3

**Summary:**

In this work, the authors propose TRAP, a benchmark designed to evaluate vulnerabilities arising from Adversarial Injections. The benchmark defines five modular dimensions and includes 630 agent susceptibility tests using realistic website clones. The authors report the average hijacking success rate across six evaluated models and the transferability rate between model pairs, demonstrating the presence of systemic security vulnerabilities.

**Strengths:**

- Addresses a more diverse set of vulnerability cases compared to prior work.

- Provides the interesting observation that a vulnerability detected in one model can transfer to other models.

- The LLM-manipulation category based on Cialdini’s principles is an interesting approach, rather than framing the problem purely as jailbreaking.

- Reports transfer rates between models through an extensive evaluation.

**Weaknesses:**

- Considers a hijack successful the moment a click occurs, but in the real world hijacking may unfold over multiple turns.

- Although the set of vulnerability cases is large, it is still limited to elements like buttons, links, and user-editable areas.

**Questions:**

- Is there a way to infer or estimate the risk/severity level for each vulnerability? (how large cost each vunlerability takes?)

---

> ### Author Response · Authors · 2025-11-21
> **Response to question and weaknesses**
>
> We thank you for your effort and acknowledgement of the strengths of our benchmark, especially recognising our broader coverage of vulnerability cases compared to prior work and finding our approach with Cialdini’s principles interesting. Below we cover the concerns mentioned in weaknesses and answer your question. As some of these points were also raised by other reviewers, we provide the same response for completeness and clarity.
>
> > **W1:** Considers a hijack successful the moment a click occurs, but in the real world hijacking may unfold over multiple turns.
>
> Thank you for raising this concern and we are happy to provide the context of why we chose the one-click criterion. Real attackers typically use lightweight injections whose goal is to **redirect the agent to an attacker-controlled domain**, where richer and adaptive prompt-injection chaining becomes possible. Once the agent clicks, the adversary can more easily perform downstream harm, such as data exfiltration, unintended transactions, or model probing.
> We demonstrate this concretely with a short video: after clicking an injected link, the agent is redirected to a malicious site and induced to leak its model identity:
> https://drive.google.com/file/d/1ak2wNn_UiDXimTsCMMyy33If98vnhm_H/view?usp=sharing
> (Names shown in the video are fictional and used only for demonstration.)
>
> The single-click metric captures the **critical point of failure**: the moment an agent follows a malicious instruction and hands control to the adversary. Long, multi-step payloads embedded directly in webpages are unrealistic. Attackers typically cannot modify the site itself, only small user-controlled fields such as reviews, comments, or listings. Larger payloads are also more visible and more likely to be rejected.
>
> A realistic hijack does not end at the first click. In the final version, we will add an extension experiment where the injected link redirects to a fully attacker-controlled webpage. There, we run second-stage prompt injections to extract sensitive information, demonstrating how a single click enables multi-step injection chaining. We include this as a new experimental section and use it to motivate future work on systematic chaining.
>
>
>
> ---
> > **W2:** Although the set of vulnerability cases is large, it is still limited to elements like buttons, links, and user-editable areas.
>
>
> Thank you for raising this point. We focus on text-based injections (buttons and hyperlinks) for three reasons:
>
> 1. **They represent the most realistic and widely accessible attack surface.**
>    On real multi-user websites (marketplaces, social networks, job boards, listings), an adversary typically controls only user-editable content. They can insert links or buttons, but cannot introduce pop-ups, banners, modals, or custom scripts. Text-based elements therefore reflect the attack surfaces most attackers can actually use.
>
> 2. **Many richer UI elements are structurally equivalent to buttons.**
>    From the agent’s perspective, pop-ups and banners typically reduce to a clickable redirect element. We use buttons as a canonical representation of these redirection mechanisms.
>
> 3. **Image-based injections are supported but not yet scalable.**
>    While TRAP allows image injections, we did not include them in this release because (i) there is currently no scalable and consistent method for generating adversarial images for agents (to our knowledge, the most up-to-date work is [4]), and (ii) image-based evaluation incurs significantly higher computational cost. We are actively extending TRAP to support text-in-image attacks within user-editable areas (e.g., images in Staynb, our Airbnb clone).
>
>
>
> ---
> > **Q1:** Is there a way to infer or estimate the risk/severity level for each vulnerability? (how large cost each vunlerability takes?)
>
> We are not entirely certain how you define risk/severity in this context, so we address one likely interpretation below, and would appreciate clarification if this does not reflect your intent.
> If the question concerns how harmful a given vulnerability is once triggered, its severity is inherently downstream and adversary-dependent. In TRAP, we intentionally stop evaluation at the first successful redirection to an attacker-controlled website, because this step gives the adversary full flexibility over what happens next (e.g., data exfiltration, social engineering, credential harvesting, or malicious downloads). For this reason, we treat all successful hijacks as equally severe at this decision point: the subsequent harm is not a property of the injection itself, but of the adversary’s follow-up strategy after redirection.
> If you meant a different notion of risk or severity, we would be grateful for clarification and are happy to adjust our response.
>
> ---
>
> We thank you again for your careful reading and constructive feedback. If our clarifications have addressed your concerns, we kindly ask you to consider updating your score accordingly.

---

### Official Review · Reviewer_22Ey · 2025-11-01

**Soundness:** 3
**Presentation:** 3
**Contribution:** 3
**Rating:** 6
**Confidence:** 4

**Summary:**

Disclosure: Claude is used to refine this review.

This paper introduces TRAP, a benchmark for evaluating hijacking vulnerabilities in LLM-based web agents. The benchmark constructs 630 task suites on 6 cloned websites by combining 5 modular components: injection interface, human persuasion principles, LLM manipulation methods, injection location, and tailoring. Using a one-click success criterion, the authors evaluate 6 models and find an average 25% hijack success rate, ranging from 13% (GPT-5) to 43% (DeepSeek-R1). Results show buttons are 3× more effective than hyperlinks, attacks transfer across models, and light tailoring substantially increases success.

**Strengths:**

- Web agent hijacking is a real and growing threat. The focus on systematic evaluation is timely and valuable.
- The 5-component framework (interface, persuasion, manipulation, location, tailoring) enables systematic ablation studies and is extensible to new attack types.
- The paper systematically examines transferability, component effectiveness, interface types, location, and tailoring, which provides actionable insights.

**Weaknesses:**

- Only buttons and hyperlinks are tested; no images, pop-ups, audio, forms, or other realistic attack vectors. Also, one-click criterion is too simplistic, as in practice agent scaffolding can detect recover from errors (e.g., [1] discussed this in detail). Further discussion and justification of the one-click criterion are needed.
- No defenses/controls are evaluated. This limits practical applicability. It would be interesting to see if input filters / output monitors can mitigate the problem.

[1] Wu et al. Dissecting Adversarial Robustness of Multimodal LM Agents. ICLR 2025. https://arxiv.org/abs/2406.12814

**Questions:**

N/A

---

> ### Author Response · Authors · 2025-11-21
> **Comment to Weakness 1**
>
> We thank you for your effort and acknowledgement of the strengths of our benchmark, especially recognising our research as timely and valuable, extending to new attack types, and providing actionable insights. Below we cover the concerns mentioned in weaknesses. As these points were also raised by other reviewers, we provide the same response for completeness and clarity.
>
> > **W1.1:** Only buttons and hyperlinks are tested; no images, pop-ups, audio, forms, or other realistic attack vectors.
>
> Thank you for raising this point. We focus on text-based injections (buttons and hyperlinks) for three reasons:
>
> 1. **They represent the most realistic and widely accessible attack surface.**
>    On real multi-user websites (marketplaces, social networks, job boards, listings), an adversary typically controls only user-editable content. They can insert links or buttons, but cannot introduce pop-ups, banners, modals, or custom scripts. Text-based elements therefore reflect the attack surfaces most attackers can actually use.
>
> 2. **Many richer UI elements are structurally equivalent to buttons.**
>    From the agent’s perspective, pop-ups and banners typically reduce to a clickable redirect element. We use buttons as a canonical representation of these redirection mechanisms.
>
> 3. **Image-based injections are supported but not yet scalable.**
>    While TRAP allows image injections, we did not include them in this release because (i) there is currently no scalable and consistent method for generating adversarial images for agents (to our knowledge, the most up-to-date work is [4]), and (ii) image-based evaluation incurs significantly higher computational cost. We are actively extending TRAP to support text-in-image attacks within user-editable areas (e.g., images in Staynb, our Airbnb clone).
>
> ---
>
> > **W1.2:** Also, one-click criterion is too simplistic, as in practice agent scaffolding can detect recover from errors (e.g., [1] discussed this in detail). Further discussion and justification of the one-click criterion are needed. [1] Wu et al. Dissecting Adversarial Robustness of Multimodal LM Agents. ICLR 2025. https://arxiv.org/abs/2406.12814
>
> Thank you for raising this concern and we are happy to provide the context of why we chose the one-click criterion. Real attackers typically use lightweight injections whose goal is to **redirect the agent to an attacker-controlled domain**, where richer and adaptive prompt-injection chaining becomes possible. Once the agent clicks, the adversary can more easily perform downstream harm, such as data exfiltration, unintended transactions, or model probing.
> We demonstrate this concretely with a short video: after clicking an injected link, the agent is redirected to a malicious site and induced to leak its model identity:
> https://drive.google.com/file/d/1ak2wNn_UiDXimTsCMMyy33If98vnhm_H/view?usp=sharing
> (Names shown in the video are fictional and used only for demonstration.)
>
> The single-click metric captures the **critical point of failure**: the moment an agent follows a malicious instruction and hands control to the adversary. Long, multi-step payloads embedded directly in webpages are unrealistic. Attackers typically cannot modify the site itself, only small user-controlled fields such as reviews, comments, or listings. Larger payloads are also more visible and more likely to be rejected.
>
> A realistic hijack does not end at the first click. In the final version, we will add an extension experiment where the injected link redirects to a fully attacker-controlled webpage. There, we run second-stage prompt injections to extract sensitive information, demonstrating how a single click enables multi-step injection chaining. We include this as a new experimental section and use it to motivate future work on systematic chaining.

---

> ### Author Response · Authors · 2025-11-21
> **Comment to Weakness 2**
>
> > **W2:** No defenses/controls are evaluated. This limits practical applicability. It would be interesting to see if input filters / output monitors can mitigate the problem.
>
> Our primary objective in this work is to systematically expose novel and practical vulnerabilities in web agents and to provide a framework for further benchmarking, rather than to propose or evaluate defences. As in prior foundational work on adversarial attacks (e.g., [5]), clearly identifying and characterising the threat is a necessary first step before rigorous defences can be designed.
>
> We agree that mitigation evaluation is important. We are currently developing a follow-up version of TRAP focused specifically on defences. In this extension, we will evaluate several practical mitigation strategies:
>
> - **Real-time monitoring layers** at key agent entry and exit points (e.g., pre-perception, post-perception, pre-action), using lightweight classifiers and heuristic filters to detect injected elements.
> - **Cross-checking agents and automated red-teamers** that review the agent’s perceived state or planned actions before execution.
> - **Sandboxed simulation pre-runs**, where agent actions are first executed in a parallel simulated environment to detect hazardous deviations before deployment.
>
> These defences will be evaluated within the TRAP framework in terms of effectiveness, computational overhead, and remaining failure modes under controlled adversarial conditions.
>
> ---
>
>
> We thank you again for your careful reading and constructive feedback. If our clarifications have addressed these concerns, we kindly ask you to consider updating your score accordingly.
>
> ---
>
> [5] A. Kurakin, I. Goodfellow, and S. Bengio. *Adversarial examples in the physical world*, International Conference on Learning Representations (ICLR), 2017.

---

### Official Review · Reviewer_LTYp · 2025-11-01

**Soundness:** 2
**Presentation:** 3
**Contribution:** 2
**Rating:** 4
**Confidence:** 4

**Summary:**

The paper introduces TRAP, a modular benchmark for task-redirecting hijacks of web agents. TRAP composes injections from five dimensions (interface, persuasion principle, LLM manipulation method, location, tailoring), yielding 630 task–injection suites across six cloned websites built on REAL. Success is defined objectively as the agent clicking an injected hyperlink or button, avoiding LLM judges and skills-gap ambiguity.

**Strengths:**

1. Frames persuasion-driven hijacks as modular components that can be recombined and extended.

2. Clear threat setup; fixed observation modality (AXTree) to control confounders; broad, factorized analysis across persuasion and manipulation methods with transferability measurements.

3. The five-component decomposition, location diagrams, and controlled comparisons (button vs hyperlink; targeted vs non-targeted prompts; tailored vs non-tailored) make the overall story legible.

**Weaknesses:**

1. TRAP’s modularity is compelling, but the paper does not empirically contrast against prior agent-security benchmarks (e.g., AgentDojo, AgentHarm, InjecAgent, etc) on overlapping attack types to show what conclusions change when using the one-click criterion.

2. Only text injections via buttons and hyperlinks are used in the core dataset; pop-ups, banners, multimedia, and richer UI elements are out of scope.

3. Using only accessibility trees improves control, but many deployed agents rely on screenshots/DOM blends. The gap between AXTree-only susceptibility and multi-modal observation remains unquantified.

4. No evaluation of simple, possible mitigations defense on the attack.

**Questions:**

1. Could the authors further clarify the contribution over prior agent-security benchmarks such as AgentDojo, AgentHarm, and InjecAgent? I suspect hijack performance may be similar between systems that use simulated environments (as in this paper) and those that use simulated observations for email or shopping (as in AgentDojo or InjecAgent).

2. I am concerned about the assumption that current LLM agents rely on the AXTree to take actions. Has this been verified empirically? My expectation is that many agents rely on screenshots or DOM representations instead.

3. Your metric stops at a click. How does the HSR translate to downstream harm in realistic workflows (e.g., redirects, data exfiltration, unintended transactions)?

4. Some hijacks transfer broadly while others do not. What features distinguish globally transferable injections from model-specific ones?

5. As a benchmark, an initial average HSR of 25% seems low, which suggests the benchmark may not be sufficiently challenging. Consider strengthening tasks or injections to better discriminate model robustness.

---

> ### Author Response · Authors · 2025-11-21
> **Positioning TRAP vs Prior Agent-Security Benchmarks**
>
> Thank you for your feedback and for recognising the strengths of our benchmark, including the clear threat model and the modular design of TRAP.
>
> In this first response, we clarify how TRAP differs from and extends prior agent-security benchmarks by introducing a systematic, modular evaluation of web-agent hijacking. Specifically, we respond to:
>
> > **W1:** TRAP’s modularity is compelling, but the paper does not empirically contrast against prior agent-security benchmarks (e.g., AgentDojo, AgentHarm, InjecAgent, etc) on overlapping attack types to show what conclusions change when using the one-click criterion.
>
> and
>
> >  **Q1:** Could the authors further clarify the contribution over prior agent-security benchmarks such as AgentDojo, AgentHarm, and InjecAgent?
>
> Our main contribution is not simply another benchmark, but **a systematic measurement of hijack success across a five-dimensional, modular attack space**. For each benign task, we evaluate **35  injection variants** spanning persuasion, LLM manipulation, interface type, location, and tailoring - a structured modularity not present in prior work. This granularity allows us to identify which attack dimensions matter, how they interact, and when small design changes double or triple hijack success. This is crucial for agent security, where real attackers iteratively adapt strategies based on observed failures.
>
> Existing benchmarks do not offer this granularity. AgentDojo uses monolithic text injections and multi-step success criteria,introducing limited modularity only via personalisation; AgentHarm focuses on harmful user prompts rather than adversarial webpage content; InjecAgent relies on LLM-judged outcomes rather than an objective behavioural metric. TRAP differs by using **realistic website clones**, a **one-click success measure**, and a **five-dimensional modular injection space**  to expose how subtle design choices drive agent failures. We expand this comparison in the Related Work section (p. 2) and summarise it below.
>
>
> | Benchmark | How it compares to TRAP | Key Differences |
> |-----------|---------------------------|-----------------|
> | AgentDojo [1] | Both evaluate prompt injections on agents. | Uses dynamic synthetic environments instead of realistic website clones; attacks are monolithic single large prompts rather than modular; success requires multi-step attack completion which confounds refusal vs incapability as noted in TRAP Section 2; no systematic variation across persuasion principles or interface elements. |
> | AgentHarm [2] | Shared goal of exposing harmful behaviour of agents. | Focuses on malicious user prompts rather than adversarial content embedded in real webpages; no modular injection space and relies on a universal jailbreak template; evaluated in simplified agent environments instead of realistic web clones like Amazon Gmail or Calendar; no analysis of interface driven failures such as buttons vs hyperlinks locations or tailoring. |
> | InjecAgent [3] | Shared goal of using prompt injection for agent misuse. | Evaluations rely on LLM judged outcomes criticised in TRAP page 2 for ambiguity and misclassification; does not isolate a one click behavioural success metric because success depends on multi-step execution; no five dimensional modular decomposition; does not test realistic web clones or human persuasion principles. |
> | TRAP (ours) | - | First modular benchmark with five independent dimensions persuasion manipulation interface location and tailoring; 630 injections systematically generated on realistic website clones including Amazon Gmail LinkedIn DoorDash Upwork and Calendar; one click success metric described in TRAP Sections 3.4 and 3.5; shows how small contextual changes such as button vs hyperlink and light tailoring can multiply hijack success; provides granular cross model transfer insight in Table 2. |
>
> ---
> [1] E. Debenedetti, J. Zhang, M. Balunović, L. Beurer-Kellner, M. Fischer, F. Tramèr.
> *AgentDojo: A Dynamic Environment to Evaluate Prompt Injection Attacks and Defenses for LLM Agents.*
> Advances in Neural Information Processing Systems (NeurIPS), 2024.
>
> [2] M. Andriushchenko, A. Souly, M. Dziemian, D. Dueñas, M. Lin, J. Wang, D. Hendrycks, A. Zou, Z. Kolter, M. Fredrikson, E. Winsor, J. Wynne, Y. Gal, X. Davies.
> *AgentHarm: A Benchmark for Measuring Harmfulness of LLM Agents.*
> International Conference on Learning Representations (ICLR), 2025.
>
> [3] Q. Zhan, Z. Liang, Z. Ying, D. Kang.
> *InjecAgent: Benchmarking Indirect Prompt Injections in Tool-Integrated Large Language Model Agents.*
> Proceedings of the Association for Computational Linguistics (ACL), 2024.

---

> > ### Author Response · Authors · 2025-11-21
> > **Methodological Design Choices**
> >
> > In this comment, we address your concerns about our methodological choices, explaining why we focus on text-based injections and AXTree observations, and how these decisions improve realism, control, and reproducibility without sacrificing generality.
> >
> > > **W2:** Only text injections via buttons and hyperlinks are used in the core dataset; pop-ups, banners, multimedia, and richer UI elements are out of scope.
> >
> > Thank you for raising this point. We have also addressed it in our response to reviewers 4gAY and 22Ey, and reproduce the explanation here for completeness.
> >
> > We focus on text-based injections (buttons and hyperlinks) for three reasons:
> >
> > 1. **They represent the most realistic and widely accessible attack surface.**
> >    On real multi-user websites (marketplaces, social networks, job boards, listings), an adversary typically controls only user-editable content. They can insert links or buttons, but cannot introduce pop-ups, banners, modals, or custom scripts. Text-based elements therefore reflect the attack surfaces most attackers can actually use.
> >
> > 2. **Many richer UI elements are structurally equivalent to buttons.**
> >    From the agent’s perspective, pop-ups and banners typically reduce to a clickable redirect element. We use buttons as a canonical representation of these redirection mechanisms.
> >
> > 3. **Image-based injections are supported but not yet scalable.**
> >    While TRAP allows image injections, we did not include them in this release because (i) there is currently no scalable and consistent method for generating adversarial images for agents (to our knowledge, the most up-to-date work is [4]), and (ii) image-based evaluation incurs significantly higher computational cost. We are actively extending TRAP to support text-in-image attacks within user-editable areas (e.g., images in Staynb, clone of Airbnb).
> >
> > ---
> >
> > > **W3:** Using only accessibility trees improves control, but many deployed agents rely on screenshots/DOM blends. The gap between AXTree-only susceptibility and multi-modal observation remains unquantified.
> >
> > > **Q2:** I am concerned about the assumption that current LLM agents rely on the AXTree to take actions. Has this been verified empirically? My expectation is that many agents rely on screenshots or DOM representations instead.
> >
> > We thank you for raising this point and have added further context on our use of AXTree (p. 3).
> >
> > In our experiments, we evaluated agents across three observation modalities: DOM, AXTree, and screenshots, both individually and in combined forms. Across modalities, vulnerabilities and attack success rates remained very similar, and ablations removing screenshots or DOM did not significantly change base task completion or attack success. This suggests that AXTree alone is sufficient for our base tasks and attacks in the TRAP setup.
> >
> > **Practical limitations of including DOM and screenshots:**
> > - **Full DOM:** Often exceeds the agent’s context window, causing truncation and unstable behaviour. Since AXTree is already a structured summarisation of DOM, including both is largely redundant.
> > - **Screenshots:** Not supported by all models and significantly more expensive. For example, running TRAP once with GPT-5 using screenshots increases total cost to approximately 270 USD. This is computed as: 26 steps × 0.017 USD per screenshot × 630 tasks.
> >
> > For these reasons, we focus on AXTree: it provides stable structure, avoids context-window issues, is supported by all evaluated models, and keeps evaluation cost manageable. Importantly, our framework still supports all three modalities for future multimodal studies.
> >
> > ---
> >
> > [4] L. Aichberger, A. Paren, G. Li, P. Torr, Y. Gal, A. Bibi.
> > *MIP against Agent: Malicious Image Patches Hijacking Multimodal OS Agents.* Advances in Neural Information Processing Systems (NeurIPS), 2025.

---

> > > ### Author Response · Authors · 2025-11-21
> > > **Hijack Semantics and Cross-Model Transferability**
> > >
> > > In this comment, we clarify the semantics of our single-click hijack metric and present our analysis of why hijacks transfer asymmetrically across models with different robustness levels.
> > >
> > > > **Q3:** Your metric stops at a click. How does the HSR translate to downstream harm in realistic workflows (e.g., redirects, data exfiltration, unintended transactions)?
> > >
> > > Thank you for raising this question - we also address it in responses to other reviewers and reproduce the explanation here for completeness.
> > > Real attackers typically use lightweight injections whose goal is to **redirect the agent to an attacker-controlled domain**, where richer and adaptive prompt-injection chaining becomes possible. Once the agent clicks, the adversary can more easily perform downstream harm, such as data exfiltration, unintended transactions, or model probing.
> > > We demonstrate this concretely with a short video: after clicking an injected link, the agent is redirected to a malicious site and induced to leak its model identity:
> > > https://drive.google.com/file/d/1ak2wNn_UiDXimTsCMMyy33If98vnhm_H/view?usp=sharing
> > > (Names shown in the video are fictional and used only for demonstration.)
> > >
> > > The single-click metric captures the **critical point of failure**: the moment an agent follows a malicious instruction and hands control to the adversary. Long, multi-step payloads embedded directly in webpages are unrealistic. Attackers typically cannot modify the site itself, only small user-controlled fields such as reviews, comments, or listings. Larger payloads are also more visible and more likely to be rejected.
> > >
> > > A realistic hijack does not end at the first click. In the final version, we will add an extension experiment where the injected link redirects to a fully attacker-controlled webpage. There, we run second-stage prompt injections to extract sensitive information, demonstrating how a single click enables multi-step injection chaining. We include this as a new experimental section and use it to motivate future work on systematic chaining.
> > >
> > > ---
> > >
> > > > **Q4:** Some hijacks transfer broadly while others do not. What features distinguish globally transferable injections from model-specific ones?
> > >
> > > Table 1 establishes a clear strength ordering: GPT-5 is the hardest to hijack (HSR = 13%), while DeepSeek-R1 and LLaMA-4 Maverick are more vulnerable (HSR = 43% and 17%). Table 2 shows an asymmetric transfer pattern. Injections that succeed on GPT-5 transfer to other models at very high rates (e.g., 90.0% to Claude Sonnet 3.7 and 88.8% to DeepSeek-R1), while injections that succeed only on weaker models transfer poorly to stronger ones (e.g., 26.0% of DeepSeek-R1 hijacks transfer back to GPT-5).
> > >
> > > This reveals a consistent pattern: **transferability follows model robustness**. Injections that break the strongest model form an approximate superset of those that break weaker models, but not the reverse. Practically, this means an adversary need only target the most robust agent; such injections are highly likely to generalise across weaker systems.

---

> ### Author Response · Authors · 2025-11-21
> **Benchmark Difficulty, Limitations, and Defences**
>
> Finally, we respond to concerns about benchmark difficulty and the absence of defence evaluations, and outline how TRAP is calibrated today and systematically extended in our follow-up work.
>
> > **W4:** No evaluation of simple, possible mitigations or defences on the attack.
>
> We thank you for raising this point. Our primary objective in this work is to **systematically expose novel and practical vulnerabilities in web agents and to provide a framework for further benchmarking**, rather than to propose or evaluate defences. As in prior foundational work on adversarial attacks (e.g., [5]), clearly identifying and characterising the threat is a necessary first step before rigorous defences can be designed.
>
> We agree that mitigation evaluation is important. We are currently developing a follow-up version of TRAP focused specifically on defences. In this extension, we will evaluate several practical mitigation strategies:
>
> - **Real-time monitoring layers** at key agent entry and exit points (e.g., pre-perception, post-perception, pre-action), using lightweight classifiers and heuristic filters to detect injected elements.
> - **Cross-checking agents and automated red-teamers** that review the agent’s perceived state or planned actions before execution.
> - **Sandboxed simulation pre-runs**, where agent actions are first executed in a parallel simulated environment to detect hazardous deviations before deployment.
>
> These defences will be evaluated within the TRAP framework in terms of effectiveness, computational overhead, and remaining failure modes under controlled adversarial conditions.
>
> ---
>
> > **Q5:** As a benchmark, an initial average HSR of 25% seems low, which suggests the benchmark may not be sufficiently challenging. Consider strengthening tasks or injections to better discriminate model robustness.
>
> We respectfully disagree that an average HSR of 25% indicates a lack of challenge. This level is consistent with related agent-security benchmarks. For example, AgentDojo reports an average ASR of 21.54%, and InjecAgent reports 29.58% in its base setting. TRAP therefore falls squarely within the current difficulty range of state-of-the-art evaluations.
>
> Moreover, a moderate success rate is a desirable property for a benchmark. If hijacks were extremely frequent (>50%) or extremely rare (<5%), the benchmark would lose discriminatory power. An HSR in the 20-30% range provides meaningful resolution: stronger models perform better, weaker models worse, and improvements or regressions remain observable.
>
> That said, we agree that progressively increasing difficulty is important. As noted on page 2, TRAP is designed as a **modular injection framework**, allowing stronger attacks to be introduced without redesigning the benchmark. We are already extending the benchmark with additional tasks, stronger textual injections, and multimodal (image-based) injections to further raise difficulty in the next iteration.
>
> ---
>
> We thank you again for your careful reading and constructive feedback. If our clarifications have addressed these concerns, we kindly ask you to consider updating your score accordingly.
>
> ---
>
> [5] A. Kurakin, I. Goodfellow, and S. Bengio. *Adversarial examples in the physical world*, International Conference on Learning Representations (ICLR), 2017.

---

### Official Review · Reviewer_Dpz7 · 2025-11-02

**Soundness:** 2
**Presentation:** 3
**Contribution:** 2
**Rating:** 4
**Confidence:** 4

**Summary:**

This paper introduces TRAP (Task-Redirecting Agent Persuasion Benchmark), a benchmark for evaluating hijacking vulnerabilities in web-based LLM agents. The authors construct 630 task suites by combining 18 benign tasks with 35 injection templates built from five modular components: injection interface (button/hyperlink), human persuasion principles (7 Cialdini principles), LLM manipulation methods (5 types), injection location, and tailoring. The benchmark is built on REAL's cloned websites (Amazon, Gmail, Calendar, LinkedIn, DoorDash, Upwork). The key innovation is a one-click evaluation metric: hijacking success is determined when the agent clicks the injected element, avoiding ambiguity from multi-step outcomes and LLM judge bias. Evaluating six frontier models, they find an average 25% hijack success rate (ranging from 13% on GPT-5 to 43% on DeepSeek-R1), with buttons 3× more effective than hyperlinks and light tailoring increasing success by up to 5.6×.

**Strengths:**

- 630 task suites on realistic website clones to measure agent susceptibility.
- Verifiable evaluation without reliance on LLM judges.
- Interesting findings regarding the vulnerability of different models (hijack success rates ranging from 13% on GPT-5 to 43% on DeepSeek-R1).
- Using realistic clones of popular websites from REAL (Garg et al., 2025). This is important as prompt injections are a major threat for agents, and prompt injection benchmarks in realistic environments are highly needed.
- Accuracy on benign tasks is also measured and provides a baseline for agents’ capabilities.
- Showing that the considered injection templates are transferable between different LLMs is important and is a nice side contribution.
- Valuable ablation studies (e.g., hyperlinks vs. buttons).

**Weaknesses:**

- My major concern is that the benchmark has a very low number of unique tasks (only 18). A total of 630 tasks are created by using 35 injection templates (7 persuasion principles × 5 LLM manipulation methods). The benchmark would be more useful with a larger number of unique tasks (say, at least 50 or better 100). Varying injection templates is less interesting, since they shouldn’t be assumed fixed (see the discussion on adaptive attacks in [Jailbreaking Leading Safety-Aligned LLMs with Simple Adaptive Attacks](https://arxiv.org/abs/2404.02151) and [The Attacker Moves Second: Stronger Adaptive Attacks Bypass Defenses Against Llm Jailbreaks and Prompt Injections](https://arxiv.org/abs/2510.09023)).
- *“We introduce a single, unambiguous success criterion: whether the agent clicks the injected element.”* - This makes the tasks easy to grade, but it’s also a weakness of the benchmark. Realistic hijacks typically require the agent to perform multiple steps.

Minor points:
- “Prompt injections” seems to be a much more established name compared to “hijacks”. I wonder why the authors seem to strongly prefer “hijacks”. When reading the paper, it was not clear to me if there is some substantial difference between them, but it seems like they refer to the same behavior.
- *“This creates ambiguity: if an agent starts to follow a malicious instruction but fails to complete it, is that a skill gap or a true refusal?”* - Note that these two things are easy to disentangle since one can measure refusals directly (e.g., as done in AgentHarm), which is a straightforward task for an LLM judge.

**Questions:**

No questions.

---

> ### Author Response · Authors · 2025-11-21
> **Comment addressing weaknesses**
>
> Thank you for your effort and acknowledgement of the strengths of our benchmark, including recognising the value of our ablation studies, verifiable evaluation without reliance on LLM judges and the interesting findings regarding the vulnerability of different models. Below we cover the concerns mentioned in weaknesses.
>
> > **W1:** My major concern is that the benchmark has a very low number of unique tasks (only 18). A total of 630 tasks are created by using 35 injection templates (7 persuasion principles × 5 LLM manipulation methods). The benchmark would be more useful with a larger number of unique tasks (say, at least 50 or better 100). Varying injection templates is less interesting, since they shouldn’t be assumed fixed (see the discussion on adaptive attacks in Jailbreaking Leading Safety-Aligned LLMs with Simple Adaptive Attacks and The Attacker Moves Second: Stronger Adaptive Attacks Bypass Defenses Against Llm Jailbreaks and Prompt Injections).
>
> We appreciate your concern and agree that scaling to more underlying tasks is a natural next step. However, the choice to use 18 unique tasks was deliberate, not a shortcut.
>
> We selected 18 tasks that collectively cover the core spectrum of realistic web-agent behaviours, while remaining computationally feasible to run at scale. Evaluating autonomous agents is still expensive and brittle; a larger task set would significantly reduce reproducibility and accessibility for other labs.
>
> Instead of inflating the number of base tasks, we found that **increasing diversity in injection templates captures performance variance more effectively**. The 35 templates produce qualitatively different attack realisations on the same benign task, and empirically we observe that variation in persuasion and manipulation style contributes more to model performance differences than variation in the base task alone. This design allows systematic control over the attack space, rather than shallow dataset expansion.
> Additionally, TRAP is not a static benchmark but an **extensible framework**. We intentionally provide a compact but well-covered task foundation, together with infrastructure for easily adding new tasks and new injection sets. This makes the benchmark adaptable as models and attack strategies evolve, without hard-coding relevance to today’s systems.
>
> In this sense, our contribution goes beyond the current set of 18 tasks and 35 templates, and focuses on providing a **scalable benchmarking pipeline** that others can extend without prohibitive cost.
>
> ---
>
> > **W2:** We introduce a single, unambiguous success criterion: whether the agent clicks the injected element.” - This makes the tasks easy to grade, but it’s also a weakness of the benchmark. Realistic hijacks typically require the agent to perform multiple steps.
>
> Thank you for your comment regarding the single-step injection - we addressed this aspect also to other reviewers and provide the same explanation here for completeness.
>
> Thank you for raising this concern - as this point was also raised by other reviewers, we provide the same response for completeness and clarity.
> Real attackers typically use lightweight injections whose goal is to **redirect the agent to an attacker-controlled domain**, where richer and adaptive prompt-injection chaining becomes possible. Once the agent clicks, the adversary can more easily perform downstream harm, such as data exfiltration, unintended transactions, or model probing.
> We demonstrate this concretely with a short video: after clicking an injected link, the agent is redirected to a malicious site and induced to leak its model identity:
> https://drive.google.com/file/d/1ak2wNn_UiDXimTsCMMyy33If98vnhm_H/view?usp=sharing
> (Names shown in the video are fictional and used only for demonstration.)
>
> The single-click metric captures the **critical point of failure**: the moment an agent follows a malicious instruction and hands control to the adversary. Long, multi-step payloads embedded directly in webpages are unrealistic. Attackers typically cannot modify the site itself, only small user-controlled fields such as reviews, comments, or listings. Larger payloads are also more visible and more likely to be rejected.
>
> A realistic hijack does not end at the first click. In the final version of the paper, we will add an extension experiment where the injected link redirects to a fully attacker-controlled webpage. There, we run second-stage prompt injections to extract sensitive information, demonstrating how a single click enables multi-step injection chaining. We include this as a new experimental section and use it to motivate future work on systematic chaining.

---

> > ### Author Response · Authors · 2025-11-21
> > **Comment addressing minor points**
> >
> > The comment below is aiming to cover the minor points:
> > > **MP1:**“Prompt injections” seems to be a much more established name compared to “hijacks”. I wonder why the authors seem to strongly prefer “hijacks”. When reading the paper, it was not clear to me if there is some substantial difference between them, but it seems like they refer to the same behavior.
> >
> > Thank you for pointing this out, we have been using both interchangeably and have changed it to now to a consistent form of ‘prompt injections’ in the improved form of the paper.
> >
> > ---
> >
> > > **MP2:** “This creates ambiguity: if an agent starts to follow a malicious instruction but fails to complete it, is that a skill gap or a true refusal?” - Note that these two things are easy to disentangle since one can measure refusals directly (e.g., as done in AgentHarm), which is a straightforward task for an LLM judge.
> >
> > Thank you for bringing this part to our attention. Our aim in pointing out the ambiguity was based on cases from papers such as [1] Table 1, where the agent refuses to conduct an action, but then still proceeds with completing it, making the refusal not sufficient. We have now improved the writing on page 1 to make our reasoning clearer.
> >
> > ---
> >
> > We thank you again for your careful reading and constructive feedback. If our clarifications have addressed these concerns, we kindly ask you to consider updating your score accordingly.
> >
> > ---
> >
> > [1] P.Kumar, E.Lau, S.Vijayakumar, T.Trinh, Scale Red Team, E.Chang, V.Robinson, S.Hendryx, S.Zhou, M.Fredrikson, S.Yue, Z.Wang; *Refusal-Trained LLMs Are Easily Jailbroken As Browser Agents*, International Conference on Learning Representations (ICLR), 2024

---

### Author Response · Authors · 2025-11-27
**Request for response to rebuttal**

Dear Reviewers,

We hope this message finds you well. As the discussion period is nearing its end with less than four working days remaining, we wanted to ensure we have addressed all your concerns satisfactorily. Could you kindly review our rebuttal and give us feedback? If there are any additional points you'd like us to consider, please let us know. Your insights are valuable to us, and we are eager to address any remaining issues to improve our work.

Below we summarise the changes we have made to our paper:

- We expanded the explanation of why a single-click captures the critical security failure point, with clearer threat modelling and stronger justification.
- We standardised terminology, consistently using prompt injection instead of mixing it with hijack.
- We added a comparison of our benchmark with AgentDojo, AgentHarm, and InjecAgent to contextualise their structure and reported difficulty level.
- We clarified why we use 18 tasks and emphasised our contribution as a modular, extensible benchmarking pipeline rather than a static dataset.
- We clearly justify the scope limitation to text-based injections and added a roadmap for future multimodal (image-based) injections.
- We added the explanation for our AXTree choice over DOM and screenshots.
- We clarified and refined this part of the introduction on refusal vs. incapability discussion.
- We polished the text to make it clearer to read.

Thank you for the comments on the single-click success criterion (point 1). Since this point was raised by multiple reviewers, we provide a unified response to address this concern:

We agree that real hijacks often involve multiple steps. However, in reality attackers often start with a minimal perturbation whose only goal is to trigger a click and move the agent to an attacker-controlled domain. That click is the critical failure point; once the agent leaves the trusted context, richer prompt-injection chaining becomes possible, enabling downstream harm (e.g., data exfiltration, unintended actions, model probing). We illustrate this with a short video where a single click enables a second-stage attack:
https://drive.google.com/file/d/1piPZyE6TQpc-Am4TMLHDMJOALiY0XsM8/view?usp=sharing

Apart from that, single-click failures are not always benign in themselves. In practice, one-click vulnerabilities regularly lead to serious compromise. For example, “Cometjacking” shows how a single click can hijack an AI browser assistant:
https://layerxsecurity.com/blog/cometjacking-how-one-click-can-turn-perplexitys-comet-ai-browser-against-you/
Similarly, recent high-severity Google Chrome vulnerabilities (e.g., CVE-2025-5419 and CVE-2025-4664) demonstrate how a single interaction with a crafted page can enable remote exploitation or cross-origin data leakage:
https://nvd.nist.gov/vuln/detail/CVE-2025-5419
https://nvd.nist.gov/vuln/detail/CVE-2025-4664

Our metric therefore isolates the first critical breach: the moment the agent follows a malicious instruction and hands control to the adversary. Long, multi-step payloads embedded directly into benign websites are often unrealistic, as attackers typically control only small user-generated fields and larger payloads are more detectable.

To demonstrate that a single-click can be chained into a multi-stage attack, we will add an extension experiment where injected links redirect agents to attacker-controlled pages, and demonstrate second-stage prompt-injection chaining and downstream harm. We keep the one-click criterion for its clarity and reproducibility, while demonstrating this can be extended to follow-on attacks.

---

### Author Response · Authors · 2025-12-02
**Summary for AC**

**Dear Area Chair,**

We are writing to summarise the state of our submission, *It’s a TRAP! Task-Redirecting Agent Persuasion Benchmark for web agents*, following the recent technical issues and score reset. We understand you are newly assigned to our paper, therefore we provide a brief overview of the review process and the key revisions made during rebuttal.

Our paper introduces *TRAP*, a reproducible benchmark for evaluating how persuasion techniques misguide autonomous web agents in realistic settings. It defines a five-dimensional modular attack space of 630 injections across persuasion methods and interface types, enabling systematic analysis of how these factors shape agent behaviour and reliability. We also provide an extensible framework that supports integrating new attacks and evaluating agents on high-fidelity website clones.

Reviewers consistently praised how timely and novel this research is, with reviewer 22Ey noting *“Web agent hijacking is a real and growing threat. The focus on systematic evaluation is timely and valuable.”*, reviewer 4gAY acknowledging that TRAP *“Addresses a more diverse set of vulnerability cases compared to prior work.“* and reviewer LTYp appreciating *“Clear threat setup”*,

The reviewers provided thoughtful and constructive feedback, which we diligently addressed through new analysis and clarifications to the manuscript.

Our key actions in response to their feedback included:
- **Benchmark alignment and positioning:** We standardised terminology to match established benchmarks and added a clear comparison with AgentDojo, AgentHarm, and InjectAgent to contextualise TRAP’s structure and reported difficulty.
- **Clarifying scope and design choices:** We justified the focus on text-based injections, explained our choice of AXTree over DOM/screenshots, and motivated the use of 18 tasks, emphasising TRAP as a modular, extensible pipeline rather than a static dataset.
- **Strengthening the single-click justification:** We clarified why a single click represents the critical breach point: real attackers typically use minimal perturbations to redirect agents off-site, after which multi-step prompt-injection chaining becomes easy. This behaviour is well-supported by real incidents (e.g., Cometjacking, high-severity Chrome CVEs). To address concerns about multi-step realism, we will add an extension experiment demonstrating how a single click escalates into a second-stage attack on an attacker-controlled page.

Here’s a brief summary of the feedback from each reviewer and our responses. Unfortunately we did not have an opportunity to hear back from the reviewers before the review rollback.

**Reviewer 4gAY:**

- **Praised:** Broader vulnerability coverage, clear cross-model transfer, effective use of Cialdini principles, and thorough transferability analysis.

- **Concerns:** Single-click metric may miss multi-step attacks; scope limited to buttons, links, and editable text.

- **Addressed by:** Strengthened justification for the single-click breach point with added chaining experiment, and clarified why text elements reflect the realistic attacker-controlled surface.



**Reviewer Dpz7:**

- **Praised:** Systematic 630-task suite on realistic clones, objective evaluation, clear vulnerability findings, strong transferability, and informative ablations.

- **Concerns:** Only 18 base tasks (preference for more), and the single-click metric may miss multi-step hijacks.

- **Addressed by:** Explained deliberate task count and value of injection diversity, and justified the single-click breach point with an added redirection–chaining experiment.


**Reviewer LTYp:**

- **Praised:** Clear modular framing, solid threat model with controlled AXTree setup, comprehensive factorised analysis, and strong interface/location/tailoring comparisons.

- **Concerns:** Limited contrast with prior benchmarks; text-only injections; AXTree-only modality; no mitigation evaluation.

- **Addressed by:** Added explicit benchmark comparison, clarified realism of text-based and AXTree design, summarised multimodal findings, and outlined defence-focused extensions.



**Reviewer 22Ey:**

- **Praised:** Timely and systematic evaluation, flexible 5-component framework, and clear analyses of transferability and component effects.

- **Concerns:** Text-only injections; single-click metric may oversimplify; no defence or monitoring evaluation.

- **Addressed by:** Clarified attacker-surface realism of text UI, expanded single-click justification with chaining experiment, and outlined forthcoming defence evaluations.

We believe the revisions made during the rebuttal period have substantially strengthened the work, resulting in a clearer and more robust contribution. We are confident that the updated manuscript addresses the reviewers’ concerns effectively. Thank you for your time and consideration. We remain happy to provide any further clarification.

Sincerely,

Authors

---

### Meta-Review · Area_Chair_zMSZ · 2025-12-03

**Summary:**

This paper introduces TRAP, a timely and new benchmark for evaluating prompt injection vulnerabilities in web-based LLM agents. Its main contribution is a new modular framework that dissects attacks across five dimensions, generating 630 test scenarios on high-fidelity website clones. The benchmark also introduces a new single-click success metric. The authors argue that it provides an objective, reproducible measurement of the initial security breach.
Their experiments across six frontier models reveal significant and systemic vulnerabilities, with an average 25% attack success rate and valuable insights into factors like interface design and attack transferability.

**Reviewer Concerns:**

Initially, the reviewers raised concerns about the usage of the "single-click success metric", and dismissed the comparison to existing methods, limited injection types, and lack of screenshots/DOM blends. The authors provided clarifications, which are mostly reasonable.

I also share the major concern (and my only concern) raised in these reviews regarding the justification of the "single-click success metric". All experiments are solely based on this metric, and the choice is justified in the rebuttal by video and anecdotal examples. I think formal justifications, like comparative analysis or analysis of real-world incidents, would largely improve the rigor of this paper.

The paper addresses a timely topic and the modular framework is well-designed, but the unresolved concern about the core evaluation metric represents a significant gap that undermines confidence in the benchmark's validity. THus, I recommend rejection.

**Reviewer Scores:**

Dpz7: 4 -> 4/6 — The reviewer's final score hinges on whether they found the rebuttal's video and real-world examples sufficient to justify the single-click metric; if persuaded, the score would rise to 6, but if they deemed the justification still incomplete, it could remain at 4.
LTYp: 4 -> 4/6 — The reviewer's concerns about benchmark comparison and design choices (AXTree, text-only injections) were addressed with detailed justifications and a comparison table. However, the lack of defense evaluation and the single-click metric concern may prevent a full increase to 6
22Ey: 6 -> 6 — The initial positive assessment was reinforced by reasonable clarifications on the metric and scope, with no new major issues arising.
4gAY: 6 -> 6 — The favorable view was sustained as the rebuttal adequately addressed the noted concerns about the metric and attack surface.

---

### Decision · Program_Chairs · 2026-01-26

Reject